# Coastal Resilience Against Storm Surge from Tropical Cyclones

**Robert Mendelsohn * and Liang Zheng**

School of the Environment, Yale University, 195 Prospect Street, New Haven, CT 06511, USA;
liang.zheng@yale.edu

* Correspondence: robert.mendelsohn@yale.edu; Tel.: +1-203-432-5128

**Abstract:** It is well known that seawalls are effective at stopping common storm surges in urban areas. This paper examines whether seawalls should be built to withstand the storm surge from a major tropical cyclone. We estimate the extra cost of building the wall tall enough to stop such surges and the extra flood benefit of this additional height. We estimate the surge probability distribution from six tidal stations spread along the Atlantic seaboard of the United States. We then measure how valuable the vulnerable buildings behind a 100 m wall must be to justify such a tall wall at each site. Combining information about the probability distribution of storm surge, the average elevation of protected buildings, and the damage rate at each building, we find that the value of protected buildings behind this 100 m wall must be in the hundreds of millions to justify the wall. We also examine the additional flood benefit and cost of protecting a $km^2$ of land in nearby cities at each site. The density of buildings in coastal cities in the United States are generally more than an order of magnitude too low to justify seawalls this high. Seawalls are effective, but not at stopping the surge damage from major tropical cyclones.

**Keywords:** tropical cyclone; flood damage; coastal resilience; storm surge; 100-year surge

## 1. Introduction

Tropical cyclones globally cause about USD 26 billion of damage per year and they cause USD 15 billion of damage in the United States alone [1]. This damage to buildings and infrastructure is expected to increase over time as coastal regions of the world expand their population and income. The bulk of the damage from tropical cyclones tends to come when major storms strike large metropolitan areas. The 10%-most harmful storms caused 93% of the total damage from tropical cyclones [2]. A large fraction of this damage is from storm surge. Can countries prevent this storm surge damage from occurring? Countries can readily affect future capital investments into low-lying coastline. However, should countries also protect major coastal cities by building massive storm walls to stop major hurricane surges? There are many tools to manage storm surges. This paper evaluates the economic effectiveness of building seawalls tall enough to reduce the damage from major tropical cyclones.

Spatially detailed analyses in a few selected locations suggest society should build seawalls to protect many low-lying urban areas. The expected flood benefits from these walls would exceed the cost of the walls [3–5]. The most desirable sites for such walls tend to be low lying urban areas where there are many high-valued vulnerable buildings. However, these initial economic studies suggest the optimal height of such seawalls is well below the height of the surge from a major tropical cyclone. These previous studies find that the additional cost of raising seawalls high enough to stop the surge from a major hurricane is far more than the additional expected flood benefit. However, these studies were performed at only a few selected sites.

This paper seeks to test whether these results likely apply to a wide range of sites along the Atlantic coast of the United States that have a range of housing densities and flood probability distributions. The study specifically measures the additional cost and additional expected benefit of raising seawalls high enough to protect buildings and infrastructure against powerful storm surges. The study examines six sites (Bridgeport CT, Middletown NJ, Norfolk VA, Wilmington NC, Charleston SC, and Jacksonville FL) along the United States Atlantic coast. These sites were chosen because they have National Oceanic and Atmospheric Administration (NOAA) tidal stations nearby with long tidal records. The analysis uses the predicted 100-year-storm as a proxy for the surge from a major tropical cyclone (We adopt the standard definition in the literature that a 100-year-storm surge is the surge with a 1% cumulative probability of exceedance). We justify this assumption because the maximum reported surge at every selected site was caused by a tropical cyclone (The maximum surge in Bridgeport came from Sandy in 2012, in Sandy Hook from Donna in 1960, in Norfolk from Chesapeake-Potomac in 1933, in Wilmington from Andrew in 2018, in Charleston from Hugo in 1989, and in Jacksonville from Georgia in 1898) and only major tropical cyclones have generated the surges approaching the predicted 100-year-surge level at these sites.

The paper does not measure the seawall benefit associated with reducing deaths from storm surge at each site. This omission would be a gross mistake for an urban area that is below sea level such as New Orleans, where 1100 people died from the storm surge of Katrina. For most of the eastern United States, however, the storm surge deaths from a major tropical cyclone have been less than 100 per storm [6]. These deaths are spread across the entire coastline struck by the storm, which will generally be wider than two hundred km, so that the expected deaths/km is less than 0.5. Given that the event will occur every 100 years, even with a value per statistical death of USD 6 million, the expected value of reduced fatalities would be less than USD 30,000 per km of sea wall length. Including expected deaths would not change the results of this study.

The paper does not address the effect of wind which also explains a large fraction of hurricane damage. Cities may well want to strengthen their infrastructure and buildings to withstand high winds. Finally, the paper does not address flooding from excessive precipitation. Some of the damage from tropical cyclones is due to freshwater flooding. It is likely that freshwater flooding from major tropical cyclones will have similar properties to the coastal surge evaluated in this paper. Major tropical cyclones will cause severe inland flooding, but such events will be rare. It may not be desirable to build very tall walls along entire waterways to stop the most severe inland flooding, but it may be advantageous to divert inland flooding away from high damage cities and towards low damage agricultural land and marshes.

This paper estimates the storm surge probability distribution for each site from the nearby NOAA tidal data of maximum annual surges. We use these data to determine the height of the storm surge one would see at each site for a typical 100-year-storm. We then calculate the flood depth for each surge to buildings depending on their elevation. Using a calibrated damage function from the literature, we calculate the damage as a fraction of the value of the buildings and infrastructure by elevation behind the wall. Finally, we calculate the marginal cost of building a wall that would protect against the 100-year storm. We test whether there are any sites that can justify building the last meter of height required to stop the 100-year surge, that is, whether the additional expected flood benefits justify the additional cost of building the seawall higher. The purpose of this exercise is not to determine the exact height seawalls should be in each city. It is simply an exercise to see if there are many circumstances where building a massive sea wall to stop hurricane surges is justified.

The study finds that the marginal cost of building a sea wall tall enough to stop a 100-year-storm averages about 33 times the expected marginal benefit. It is not a good strategy to build walls as high as this. The expected marginal flood benefit is low because the probability of these powerful storms striking a specific place is rare. The marginal cost is high because the walls must be tall enough and sturdy enough to stop the immense power of the surge from a major tropical cyclone. There are few places in the United States where the potential value of property behind a 100-year-wall is sufficiently

dense. There will be exceptions to this rule, where a small wall protects a large area or where a city may be dense enough or below sea level such as Manhattan or New Orleans, respectively. These exceptions deserve more careful study. However, for most cities, the building density is not nearly high enough to justify protecting against the storm surge of major hurricanes. The results suggest that massive seawalls are generally not an effective tool to reduce the damage from major tropical cyclone storm surges.

## 2. Results

Figure 1 presents the estimated storm surge probability distributions for each site in the study. Wilmington NC and Norfolk VA have relatively low expected storm surge heights, whereas Bridgeport CT has a relatively high expected storm surge height compared to the other sites. The tail end of these distributions describes where the 100-year surge will lie. The parameters of the GEV distribution function and goodness-of-fit figures are show in the Supplementary Materials. The more extended the tail of the distribution in Figure 1, the higher the 100-year surge will be. The figure suggests that the height of the 100-year surge will be the lowest in Wilmington (1.69 m) and the highest in Bridgeport (3.25 m) amongst these sites. The 100-year storm surge has a 1% cumulative probability of being exceeded, as shown in Figure 2.

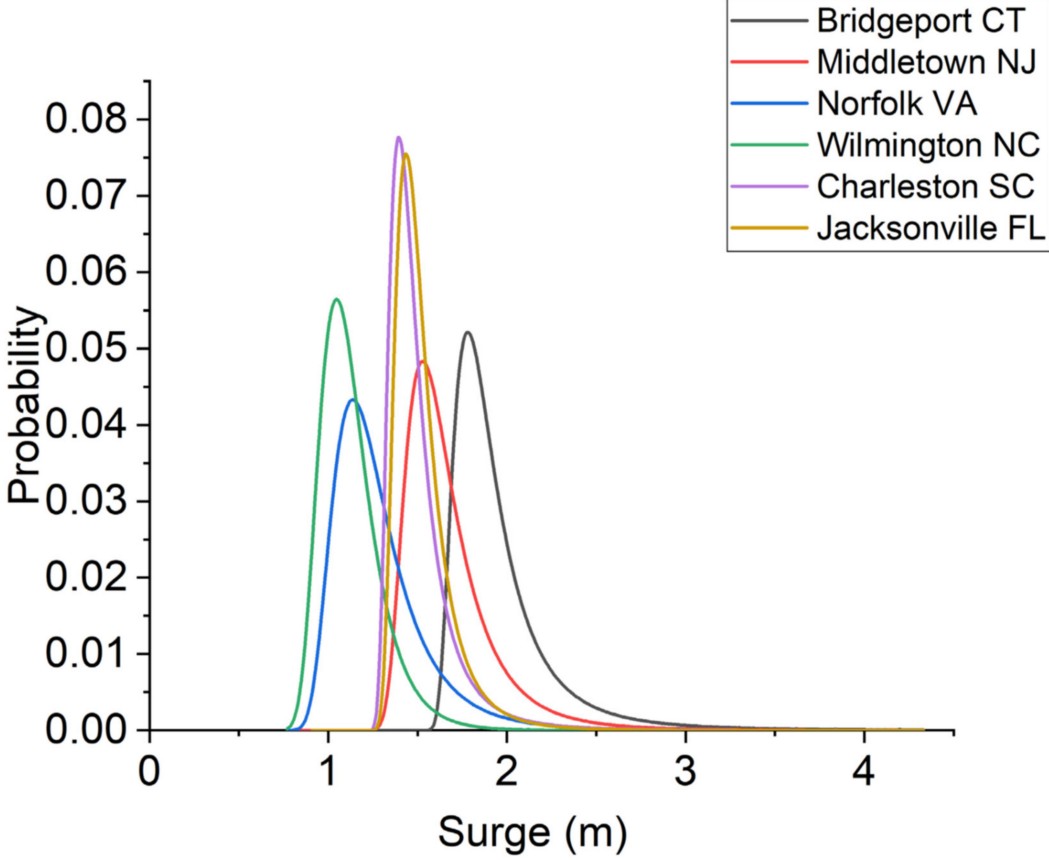

**Figure 1.** Probability distribution of storm surge by site. Source: Author estimation using data from [7] and a Generalized Extreme Value function.

In our initial analysis, we imagine a hypothetical wall that is 100 m long built along the coast at MHHW (Mean High High Water). We calculate the marginal cost and marginal expected flood benefit of building the wall high enough to stop a 100-year storm surge. The required height, shown in Table 1 for each site, is calculated from the surge probability distribution at each site. The marginal cost to reach this desired height is also shown in Table 1. The marginal cost of making a 100 m wall a meter higher averages slightly more than USD 95,000 across the six sites. The higher the wall, storm surge height

minus MHHW, the greater is the marginal cost. The next column in Table 1 shows the rate of expected marginal damage per million dollars of vulnerable building value per meter of height. The expected marginal damage averages USD 273 per million dollars of protected buildings. The higher the storm surge, the lower is the expected marginal damage per building value because the tail of the distribution is thinner. The final column presents the value of the vulnerable buildings that must lie behind the 100-m wall to justify the height. On average, the value of the vulnerable structures behind the wall must be worth USD 320 million to justify the wall that stops a 100-year storm. There are not many places along the US coast that have this much building value lying behind a 100m wall. In the case of Bridgeport, the building value must be USD 633 million because both the marginal cost is high and the marginal damage per value of vulnerable buildings is low.

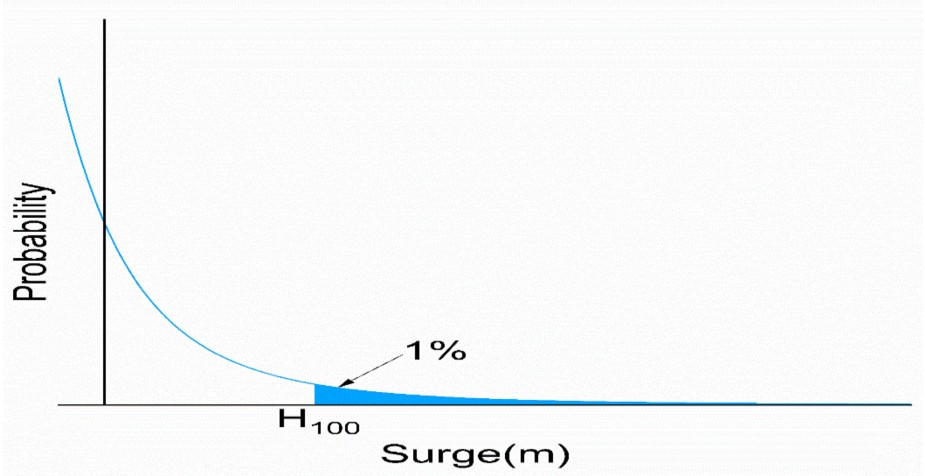

**Figure 2.** Height and Probability of 100-Year-Surge.

**Table 1.** Minimum Value of Protected Buildings That Justify Wall Height to Stop a 100-Year Storm Surge (Wall length 100 m).

| City | Height (m) | Marginal Cost (USD) | Marginal Damage/ Value (USD/million USD) | Required Value of Protected Buildings (USD Million) |
|---|---|---|---|---|
| Bridgeport, CT | 2.2 | 126,529 | 200 | 633 |
| Sandy Hook, NJ | 1.9 | 110,640 | 340 | 325 |
| Norfolk, VA | 1.9 | 109,774 | 360 | 305 |
| Wilmington, NC | 1.1 | 60,664 | 480 | 126 |
| Charleston, SC | 1.5 | 87,241 | 280 | 312 |
| Jacksonville, FL | 1.3 | 77,419 | 360 | 215 |

Source: Author calculation.

In our second analysis, we directly compare the marginal cost with the marginal benefit of a wall that stops a 100-year storm surge for a 1 km² average piece of land in each city. We assume the land has the average slope of the coastal eastern United States of 1/500 [8]. We calculate the depth of protection behind each wall. Taller walls lead to protection further away from the wall. Given the depth of protection, we calculate how long the wall would have to be along the coast to protect 1 km² of land. As shown in Table 2, depending on the height of the wall, the wall would have to be between 627 to 976 m long. The expected flood damage rate per million dollars of building value is the average across buildings whose elevation ranges from MHHW to 1 meter above the top of the wall. We then use the density and average value of the buildings in the city in order to calculate the marginal damage for that city.

**Table 2.** Marginal cost versus expected marginal benefit to protect against 100-year storm. (Wall protects 1 km$^2$ area).

| City | Depth (m) | Length (m) | Height (m) | Marginal Cost (USD) | Marginal Benefit (USD) |
|---|---|---|---|---|---|
| Bridgeport CT | 1595 | 627 | 2.2 | 793,285 | 33,256 |
| Middletown NJ | 1458 | 686 | 1.9 | 759,108 | 10,183 |
| Norfolk VA | 1450 | 690 | 1.9 | 757,062 | 27,788 |
| Wilmington NC | 1025 | 976 | 1.1 | 591,844 | 37,633 |
| Charleston SC | 1255 | 797 | 1.5 | 695,147 | 9557 |
| Jacksonville FL | 1170 | 855 | 1.3 | 661,701 | 8294 |

Source: Author calculation.

The marginal cost of making the wall tall enough to stop a 100-year storm surge averages USD 710,000. In contrast, the expected marginal damage avoided from these taller walls average only USD 21,000. The average marginal cost is 33 times the marginal damage. There is no economic justification for building the wall to be tall enough to stop the 100-year storm. A lower wall may well be justified because the marginal cost might be low enough and the marginal damage might be high enough. However, the added protection by making the wall taller does not justify the expenditure. Society gets back about three cents in flood benefits for every dollar spent making the wall taller. The 100-year storm height is not close to being justified at any of the six sites.

We then explore whether the results are robust against some of the assumed parameters of this analysis. The first robustness check tests how the results would change if the wall had to prevent a 100-year storm surge plus accumulated sea-level rise (SLR) by the end of its lifetime. We incorporate the observed historic relative rate of SLR at each site. As shown in the first column of Table 3, the average marginal cost falls 3.5% relative to the base case. The wall is taller to accommodate the additional SLR. However, the taller wall covers a greater depth and therefore a shorter length of wall is needed to protect 1 km$^2$. The marginal benefit increases slightly because the SLR increases the depth of flooding. The average damage rises by 7.5% because of SLR. Adding SLR does not change the substantial gap between marginal cost and marginal damage.

**Table 3.** Sensitivity analysis.

| City | Historic SLR MC | Historic SLR MB | Flatter Slope MC | Flatter Slope MB | Higher Probability MC | Higher Probability MD |
|---|---|---|---|---|---|---|
| Bridgeport CT | 771,518 | 35,387 | 396,643 | 33,256 | . . . | . . . |
| Middletown NJ | 728,374 | 10,870 | 379,554 | 10,183 | . . . | . . . |
| Norfolk VA | 721,960 | 29,589 | 378,531 | 27,788 | . . . | . . . |
| Wilmington NC | 570,955 | 40,986 | 295,922 | 37,633 | 591,844 | 81,972 |
| Charleston SC | 668,770 | 10,339 | 347,574 | 9557 | 695,147 | 20,677 |
| Jacksonville FL | 643,549 | 8981 | 330,850 | 8294 | 661,701 | 17,962 |

Source: Author calculation.

The second robustness check we explore is cutting the slope of the land in half. This dramatically changes the marginal cost of the wall by shortening the length of the wall by half. The marginal damage from protecting 1 km$^2$ of land does not change. With these much lower values, the marginal cost is 16 times higher than the marginal damage on average.

The third robustness check is to reduce the probability of the 100-year storm by half to capture a possible bias in the Generalized Extreme Value (GEV) function for southern sites. This doubles the expected marginal benefit of the seawall in Table 2 at each southern site. The sensitivity was not

conducted on the more northern sites because there was no indication of any possible bias. In the southern sites, the ratio of average marginal cost to average marginal damage goes from 31 in Table 2 to 15.5 in Table 3 with the higher probabilities of storm surge.

## 3. Discussion

When major tropical cyclones strike large cities, they cause enormous damage. This paper examines whether building seawalls in major cities will be an effective mechanism to reduce the storm surge damage from tropical cyclones. We focus on protecting against the 100-year storm surge as a proxy for stopping the storm surge from major tropical cycles.

The study finds that the marginal cost of raising seawall heights so that walls are both tall enough and strong enough to stop 100-year storm surges generally will outweigh the expected marginal benefit. Given this result, fair flood insurance is a less costly way to manage the damage from these rare but high damaging events. Of course, if a local entity can get an outsider to pay for their seawall, there is a private benefit to the construction. However, building sea walls high enough to stop 100-year storms makes the flood problem worse for society by making it more costly. It is a maladaptation to coastal storm surges.

The analysis does not delve into the geographic details of each local area. A careful micro approach is critical in designing exactly where to place walls to prevent flooding and how high they should optimally be [3–5]. This study steps back from this local detail to ask whether the tall walls that can stop major storm surges makes sense in general. We posit that it is not necessary to determine the exact cost and damage of walls for 100-year storms to see that it is a bad idea. Obtaining the precise details of typical cases will not change the results.

However, there could be special circumstances where a 100-year wall might make sense. The densest county in the United States is Manhattan. For example, given the risk of inundation from a major tropical cyclone at Sandy Hook, near New York City, an urban area with USD 4.1 billion of building value per $km^2$ would justify being protected from a major tropical cyclone. The value of buildings in Manhattan is USD 14.1 billion/$km^2$, so Manhattan may be an exception. However, only a fraction of Manhattan is low enough to be vulnerable, therefore one would have to undertake a very careful spatial analysis of the elevation and value of buildings throughout Manhattan to determine the optimal seawall heights. Another possible exception is New Orleans, which is largely below sea-level. A spatially detailed analysis of New Orleans could determine the value of buildings by elevation and the length of desired walls and their optimal height. Cities below sea level have much higher flooding levels given any specific storm surge and therefore have more damage per building and more lives lost when the city is flooded. The damage wreaked by Hurricane Katrina in 2005 is a perfect example. Entire countries which are largely below sea level, such as the Netherlands, are another important exception. The key for the world is to treat these places as exceptions that deserve careful study rather than as representative models for what the rest of the world must carry out.

The paper analyzes how best to tackle the problem given the current conditions. Over many decades, the conditions may change, and decisions would have to be reassessed. For example, by the end of the century, with future sea-level rise and possible changes in storm surge distributions, the literature suggests more seawalls and taller walls would be needed [9–12]. However, this study makes clear that building those future walls today is premature. How exactly changes in sea-level rates and changes in storm climatology should alter future seawalls is a subject that needs more investigation.

## 4. Materials and Methods

The basic theory behind building a defensive coastal wall is to maximize the net benefits: Benefit, *B(H)*, minus Cost, *C(H-$A_W$)*:

$$\max_{H} B(H) - C(H - A_W)) \tag{1}$$

where $H$ is the elevation of the top of the wall and $A_W$ is the elevation of the land upon which the wall is built. Taking the derivative of Equation (1) with respect to H yields the formula for the optimal wall height:

$$MC(H^* - A_w) = MB(H^*) \tag{2}$$

The optimal height of the wall, $H^*$, satisfies Equation (2). The marginal cost, $MC(H^* - A_w)$, is equal to the marginal benefit, *MB(H\*)*. If the wall is too high and exceeds $H^*$, the net benefits of the wall start to fall. The additional cost becomes greater than the additional benefit. Society is better off building the wall at $H^*$ than a taller wall.

Looking more carefully at what lies underneath the marginal benefit of seawall protection, one must measure the expected marginal benefit at each storm surge height, h. The expected marginal benefit is the product of the probability of that surge, *π(h)*, times the consequence of that storm surge. Given flood damages from past major storms, flood damage to buildings is about half of all the flood damage [13]. We consequently measure the total flood benefits as twice the flood benefits to the buildings.

The annual expected damage avoided from the last meter of sea wall height depends on the probability of the storm surge that is as high as the wall, $\pi_t(h)$, times the consequence when that storm surge occurs. The consequence is measured by a damage function that depends on the depth of flooding (surge height, $h$, minus the elevation, $A_i$) for each building $i$, times the value of the building, $V_i$.

$$E[MB(H)] = \pi_t(h)\left(\sum D_i(h - A_i)V_i\right) \tag{3}$$

This study is specifically interested in determining whether the annual marginal benefit of building a wall to stop the 100-year storm surge, $h_{100}$, is equal to or greater than the annual marginal cost.

$$E[MB(h_{100})] \equiv \pi_t(h_{100})\left(\sum D_i(h_{100} - A_i)V_i\right) \geq MC(H_{100} - A_w) \tag{4}$$

Assuming that the value of buildings is evenly distributed between the elevation where the wall is built ($A_w$) and the elevation where protection ends ($A_i = 1 + h_{100}$), it is relatively straightforward to determine an average damage per unit of value, $D(h_{100} - A^*)$, for all the vulnerable property protected by the wall. If the distribution of buildings is not evenly distributed, one can still determine an average rate of damage per unit of vulnerable building, but it will depend on the distribution of building elevations. Equation (4) becomes:

$$\pi_t(h_{100})(D(h_{100} - A^*)\left(\sum V_i\right) \geq MC(H_{100} - A_w) \tag{5}$$

Manipulating Equation (5), one can determine the minimum aggregate value of the vulnerable buildings, $\sum V_i$, that justify a 100-year wall:

$$\sum V_i = MC(h_{100} - A_w) / [\pi_t(h_{100})D(h_{100} - A^*)] \tag{6}$$

The aggregate value must be higher if the marginal cost of the wall is higher and if the probability of the storm times the average rate of damage per value of vulnerable building is smaller.

The probability of storm surge was estimated from tidal station data [7]. We utilize the maximum observed tide each year to calculate a Generalized Extreme Value (GEV) function for each city. The maximum annual tide was used to assure that the function captures the largest observed surges in each dataset because these are the surges from major tropical cyclones that cause all the biggest damage. Quarterly maximum data lead to a GEV that tends to underestimate the largest surges. Looking at the figures in the Supplementary Materials for each site compares the actual and predicted value for the largest observed storms. These figures reveal that the model overestimated the return period for the largest storm surge at the three most southern sites: Wilmington, Charleston, and Jacksonville. The actual frequency of these large storm surges appears to be more often than the GEV predicts.

We return to examine the potential effect of using the actual frequency versus the predicted probability of these storms in the sensitivity analysis. The data at each site have a minimum number of observations of 50 years although four of the sites have over a hundred years of data. The data were first detrended to remove the predicted influence of sea level rise [14]. The GEV parameters are presented in the Supplementary Materials, Table S1 for each site.

Flood depth to each building is assumed to be the difference between surge height and the elevation of the ground upon which a building is built. The damage function used in this analysis assumes the rate of damage per unit value of a building with a basement is a linear function of flood depth which is equal to zero at 1 m below ground and destroys the building completely at 8 m [15].

The construction cost of building a wall to stop storm surge is proportional to its length, *L*. Because the wall is effectively a triangle with a wide base, the construction cost of the wall increases with the square of its height (*H*) as first suggested by [9]:

$$Construction\ Cost = B\,H^2 L \tag{7}$$

This study relies on an updated cost estimate of B of USD 5200 for a 1 m high hardened wall that is 1 m in length. We assume a municipal bond that lasts the lifetime of the project, *T*, covers the construction cost. The coastal wall is expected to last 30 years after which it will have to be replaced [16]. The annual bond payment would also depend on the bond interest rate r. The municipal bond interest rate is three percent [17]. In addition, there would be an annual maintenance cost equal to 0.005 times the capital cost [16]).

$$Annual\ Cost = \left(0.005 + \frac{r}{1 - e^{-rT}}\right) BH^2 L \tag{8}$$

The annual marginal cost is the derivative of Equation (8) with respect to *H*:

$$Marginal\ Cost = \left(0.01 + \frac{2}{1 - exp^{-rT}}\right) BHL \tag{9}$$

We conduct three sensitivity analyses to test whether the results are robust against model parameter assumptions. In each case, we continue to compare the marginal cost versus the expected marginal benefit of protecting 1 km$^2$ of land against a major tropical cyclone. A typical sea wall has a 30-year life expectancy [16]. We test what would occur if relative sea level rise (SLR) continues at its historic rate at each site. We increase the height of the wall so that it would stop a 100-year storm at the end of its lifetime with 30 years of SLR. We also test what would occur if the slope of coastal land was half the average slope in the East Coast. By dividing the slope by two, a wall of a specific height would protect twice as far into the interior. In order to protect 1 km$^2$ of land, the length of the wall along the coast would be cut in half. This will lower the marginal cost in half. If the buildings are a uniform density, the expected marginal damage would be unchanged.

Armed with the marginal cost and marginal damage functions, we conduct three analyses. In all cases, we assume the potential seawall is built at Mean High High Water (MHHW). Walls at MHHW will not be subject to constant wave action which limits their maintenance cost. Many states control land below MHHW and would likely prohibit construction below this elevation.

The first analysis examines a 100 m length of wall along the coast that is assumed to protect a stretch of land behind the wall. Given an assumption that the vulnerable coastal buildings behind the wall are 0.33 m above MHHW, the analysis asks what the value of these vulnerable buildings must be to justify this wall being tall enough to stop a 100-year storm surge using Equation (5).

The second analysis concerns a wall that will protect 1 km$^2$ of land along the coast in each city. Given the height of the 100-year storm surge and the slope of land in the eastern United States (1/500) [8], this analysis first calculates the depth of land that would be protected by each wall. It then adjusts the length of the wall along the coast required to protect 1 km$^2$ of land. If the value of buildings is evenly distributed from the base of the wall, $A_w$, to the maximum elevation of affected properties, $H_w + 1$,

an average damage rate per unit value can be calculated for the 100-year storm surge. Given the average value per unit of housing and the average density in each city, one can calculate the expected marginal benefit of the wall that stops the 100-year storm. This can then be compared with the marginal cost of that wall. We proxy the density of housing units in each city with the observed density of households in that city assuming there is one household for each unit of housing.

The third sensitivity analysis examines the potential overestimation of the return period for Wilmington, Charleston, and Jacksonville. The return period for each storm height for large storms appears to be about half as short as the model is predicting. The 100-year storm appears to be occurring every 50 years. This could be a random event and does not need to be corrected. However, it is possible the GEV is assigning too low a probability to large storm surges. In this sensitivity analysis, we assign a 1/50 probability to the predicted 100-year storm and recalculate the expected marginal damage.

## 5. Conclusions

Seawalls are an effective way to reduce flooding damage in low-lying dense coastal cities. Seawalls have been successfully implemented in cities around the world. This paper examines whether seawalls could be an effective strategy to prevent the damage from major tropical cyclone storm surges. The study specifically measures the marginal cost and marginal benefit of seawalls that are high enough to stop such storm surges. If the marginal benefit is equal to or greater than the marginal cost, then such tall seawalls would be economically justified.

The study tests this hypothesis by examining a set of six cities along the Atlantic Coast that happen to have tidal stations nearby with long term measurements of storm surge. The tidal data provide a measurement of the probability distribution of extreme storm surges in each place. This allows the study to determine the likely height of a 100-year storm at each site. At every one of these sites, the largest observed storm surge has, in fact, been a named tropical cyclone. We then perform two experiments. We measure what the value of the buildings must be behind a 100 m wall that would justify the wall being tall enough to stop the 100-year storm. We measure the expected marginal benefit versus marginal cost of protecting an average 1 km$^2$ block of land at each site.

The study finds that the value of the buildings behind the 100 m wall must be worth around USD 300 million to justify a hurricane proof wall. The value is so high because powerful tropical cyclone storm surges occur very infrequently at a single location, but when they do occur, only a very expensive seawall will stop them. There are simply few low-lying coastal facilities that have this concentrated value in one small place. Facilities that are this valuable are generally built away from the coast in safer sites.

The study also finds that the marginal cost of protecting a typical 1 km$^2$ of land at each site in the study average about 33 times the marginal benefit. The extra height required to stop a 100-year storm returns about three cents in flood benefits for every dollar spent on protection. It is generally not an attractive proposition to protect American cities from major tropical cyclones using seawalls. Seawalls are too expensive to stop the storm surge from major tropical cyclones.

It is important to note that the study did not examine whether seawalls are effective at stopping more frequent smaller storm surges. There are many studies in the literature that reveal that seawalls are effective at protecting low lying dense urban areas from frequent storms. The negative results about seawalls in this paper apply only to preventing the damage from major tropical cyclones.

There may be two exceptions in the United States to the general result of this study where cities are either dense enough to justify tall walls (Manhattan) or they are below sea level (New Orleans) and are therefore more vulnerable. Only a future careful study of these two sites could determine the desirable outcome. However, these places are exceptions rather than illustrations of how best to use seawalls in general.

Development will change future protection decisions if there is far more at risk in the future along the coast. Climate change will change these future risks as the oceans rise and storm intensities increase. Seawalls will likely play an ever more important role in controlling the damage from the

additional risks of common storms. However, more research is needed to determine whether future major tropical cyclones can be stopped by seawalls or not. It may be even harder to stop future storms if they are, in fact, more powerful.

**Supplementary Materials:** The following are available online at http://www.mdpi.com/2073-4433/11/7/725/s1.

**Author Contributions:** R.M. conceived the theory and methods and wrote the report and L.Z. downloaded the data and did all the empirical work and figures. All authors have read and agreed to the published version of the manuscript.

**Funding:** This research received no external funding.

**Acknowledgments:** We thank the referees for their helpful comments.

**Conflicts of Interest:** The authors declare that they have no conflicts of interest.

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
