# Peer review of "Coastal Resilience Against Storm Surge from Tropical Cyclones"

_atmosphere, doi:10.3390/atmos11070725_

Round 1

Reviewer 1 Report

The authors have addressed many of the issues that they had with their initial submission. One point that remains is a discussion of the representativeness of the results to the entire coastal US. There are many local effects that might affect the conclusions of the paper. Additionally, the authors might want to consider that their results are based on a single 100-year storm, but if the infrastructure is built to protect from smaller events, those conditions also need to be consider in their analysis. A discussion of this might also benefit the study.

Author Response

The reviewer is concerned whether or not the results are representative. 

The last revision made clear that there are two possible exceptions in the United States where the results may not hold: New Orleans because it is below sea level and Manhattan because the density of buildings is high enough. The paper recommends both these sites be studied more carefully. 

The reviewer is also concerned we are not fairly judging the cost versus the benefit of making the seawalls tall enough to stop a major tropical cyclone.

We have consequently revised both the introduction and the abstract to make clear the analysis is comparing the additional cost to raise a wall tall enough to stop the estimated 100 year surge against the additional expected flood benefit. The additional cost is over 30 times the additional benefit.  Because this comparison is not even close to 1, more detail is not needed to rule out building walls this tall.  We agree that New Orleans and Manhattan deserve a more careful analysis because of their special circumstances before making a final judgement in these two locations.   

Reviewer 2 Report

This manuscript has been improved a lot and is acceptable for publication.

Author Response

Thank you for your earlier comments.  They were very helpful in revising the paper.  We are glad that you are pleased with the revision.  

This manuscript is a resubmission of an earlier submission. The following is a list of the peer review reports and author responses from that submission.

Round 1

Reviewer 1 Report

This paper attempt to do a cost benefit analysis for storm surges for selected cities on the east coast of the USA.

Overall I found the paper well written and suitable for publication.  The only issue is the description of the GEV analysis and reporting on some measure of the uncertainty of the 100y storm surge.

Reviewer 2 Report

This manuscript discuss whether to build seawalls to protect coastal cities against major tropical cyclone storm surges. The topic is interesting however this manuscript didn't take into consideration the importance of human life. It's better to add more factors in the manuscript to better discuss the issue related to seawalls against storm surges.

    Minor issues:

  1. It's better to move some sentences and paragraphs in "1. Introduction" to "2. Materials and Methods" and "4. Discussion".
  2. If no "conclusion" is addressed, the conntribution of the manuscript will be discounted. 百度知道    - 百度快照

Reviewer 3 Report

The authors present results of an evaluation of the benefits of seawalls as coastal protection. The method they apply is well-known and used everyday in evaluations of coastal engineering feasibility. The data they use is only partially useful to the study as they don't include many of the driving processes affecting coastal engineering design. They do not include any local effects or any future estimations. The lack of sea level rise effects is glaring. Some of the statements like "Limiting construction to MHHW also will not interfere with the normal flow of coastal rivers and streams" is also wrong and has an entire field of study in coastal engineering on the effects of this kind of projects on surrounding areas such as rivers and estuaries. Any of the projects evaluated by the U.S. Army Corps of Engineering currently in the US include many of this factors and they provide regular assessment of the benefits of this infrastructures. The authors also neglect to point out the seawalls are only one of the many tools that coastal engineers use to manage storm surge. Also, the venue for publication seems odd, when more appropriate journals that relate to storm response, coastal engineering and ocean processes are available even at MDPI. Considering all of this and many other issues with the paper, I recommend rejection.

Reviewer 4 Report

The manuscript "Coastal resilience against storm surge from tropical cyclones" builds a simple mathematical model to show whether building seawalls high enough to prevent 100-year storm surges due to tropical cyclones is cost effective in terms of building damage. Six sites along the eastern U.S. coast are used for analysis, the marginal costs for building and maintaining the seawalls are compared to the marginal benefits for saving buildings, and the conclusion is drawn that the construction of 100-year-surge-resilient seawalls is not justified, since the cost largely outweighs the benefit given the average value and density of the buildings at these sites.

While this topic is interesting to coastal policy makers, I have a few major questions on the design and assumptions of this study as detailed below. I would not recommend the publication of this manuscript unless these comments are addressed.

1. Why is marginal cost and benefit, instead of absolute cost and benefit, treated as the target here? Since the question is whether building such a seawall is worthwhile, absolute cost effectiveness seems a more proper choice to me. Otherwise more explanation is needed for the authors' current model design.

2. The analysis made several major assumptions when calculating potential building damage, including 1) value and location of buildings in affected areas are uniformly distributed; 2) land slope is assumed to be a fixed constant at all sites; 3) an implicit assumption that the protection area of a seawall is the land straightly behind it. While these assumption may not necessarily affect the final conclusion, I find some of them (in particular the third one, since the lateral flow of water could significantly change the damaging area) oversimplified, which makes the results at most a rough estimate than a reliable quantification.

3. A few clarifications are needed:
1) how exactly is the term D(H-A) calculated?
2) when detrending the annual-maximum surge data, is the same trend applied to all sites? Presumable this should be the case if the purpose of detrending is to remove overall sea level rise;
3) the land slope in the eastern U.S. is quoted at two places in the paper (method section and result section) with different numbers;
4) is there a reason why one-time building damage is compared to annual cost?